# Stakeholder Insights into Czech Performance-Based Managed Entry Agreements: Potential for Transformative Change in Pharmaceutical Access?

**DOI:** 10.3390/healthcare12010119

**Published:** 2024-01-04

**Authors:** Petra Hospodková, Pavel Karásek, Aleš Tichopád

**Affiliations:** Departement of Biomedical Technology, Faculty of Biomedical Engineering, Czech Technical University in Prague, 272 01 Kladno, Czech Republic; pavel.karasek@mensa.cz (P.K.); ales.tichopad@fbmi.cvut.cz (A.T.)

**Keywords:** performance-based managed entry agreements, stakeholder analysis, qualitative research, innovative medical technology, pharmaceutical industry, risk-sharing

## Abstract

Managed Entry Agreements (MEAs) play a pivotal role in addressing the challenges arising from escalating prices of innovative medical technologies, especially in areas like oncology, immunology, and rare diseases. Among MEAs, Performance-Based MEAs (PB MEAs) and Outcome-Based MEAs (OB MEAs) stand out as innovative strategies. This study examines the adoption of PB MEAs in the Czech Republic post a 2022 legislative change. Interviews with key stakeholders, including the Ministry of Health, pharmaceutical companies, insurers, and patient groups, were conducted to explore perceptions and challenges. Stakeholders expressed concerns about legislation completeness, data quality, transparency, and methodology. Interestingly, pharmaceutical companies were less concerned about transparency and methodology, likely due to their multinational experience. Despite legislative progress, challenges persist, especially in data infrastructure, risk-sharing perceptions, and stakeholder readiness. Addressing these issues requires collaboration between pharmaceutical companies and payers. Patient involvement, though mandated, remains limited, potentially due to a lack of awareness. This study emphasizes the need for a comprehensive transformation beyond legislation for a successful PB MEA implementation. Trust, technical infrastructure, and data availability are crucial, necessitating a holistic approach. It contributes to the global discourse on PB MEAs, stressing the adjustment of financial frameworks, embracing value-based healthcare principles, and ensuring high-quality health data metrics. A more holistic, value-based MEA approach could reshape pharmaceutical reimbursement in the future.

## 1. Introduction

Managed Entry Agreements (MEAs) epitomize innovative strategies to bridge the gap between payers and manufacturers, considering cost, risk, and pricing factors. As novel medical technologies and treatments—particularly in oncology, immunology, and rare diseases—witness escalating prices, MEAs offer a mechanism to ensure enhanced accessibility to these breakthrough interventions [1]. Klemp et al. [2] describe MEAs as structured agreements enabling access to medical technologies under stipulated conditions. This facilitates a more adaptive healthcare system capable of integrating new therapies while managing economic sustainability.

Specific types of these agreements comprise so-called performance-based MEAs (PB MEAs) or outcome-based MEAs (OB MEAs). In this regard, OB MEAs are understood as a subset of PB MEAs. In an OB MEA, the reimbursement or pricing of a drug is directly linked to the observed clinical outcomes or real-world evidence generated during the use of the treatment. These agreements often involve continuous monitoring and data collection to assess whether the treatment meets predefined effectiveness or safety criteria. If the treatment fails to achieve the desired outcomes, reimbursement terms may be adjusted or reconsidered. Such dynamic engagement reflects a shift towards more responsive and patient-centric healthcare models.

PB MEAs, as the overarching concept, consider evidence of a treatment’s benefits in terms of final results or measurable performance, like whether patients keep using the drug. These details guide decisions about whether to reimburse or continue reimbursing a specific medicine [3]. MEAs applied in real reimbursement programs reduce the consequences of making a poor coverage decision in the face of uncertain effects of a new treatment on health outcomes and/or healthcare budgets [4,5,6].

Expanding coverage for therapies that are subsequently proven ineffective, or, conversely, withholding coverage for those that are later identified as effective or cost-efficient, can result in suboptimal health outcomes and the misallocation of resources. This situation arises from either denying patients’ access to beneficial treatments or from the provision of treatments that lack efficacy [7,8]. In the long term, poor decisions can compromise the credibility of decision-making processes and cause skepticism among stakeholders and the public [7]. Both PB MEAs and OB MEAs are characterized by the need for data analysis on the efficacy of the product. The emphasis here is on ensuring that coverage and payments are directly tied to the results beneficial to the patient and, thus, the society as a whole [6,9]. There are various other arrangements known as Service-Based Agreements, comprising sponsoring by manufacturers, etc. [10,11]. 

The OECD [9] survey and public sources indicate that by 2019, MEAs were being used or had been used in at least 28 of the 41 countries that are members of the OECD and/or the European Union. Globally, the implementation of MEAs varies significantly, with each country adopting its distinctive approach and terminology. These arrangements are known by various names, including “deeds” in Australia, “special pricing arrangements”, “risk-sharing agreements”, as well as “conventions” in Belgium, and the “patient access schemes” prevalent in the United Kingdom.

Experiences with PB MEAs are still limited. Little information is available on how successful payers have been so far in using PB MEAs to meet their stated objectives. This is because few countries have formally evaluated their experience with PB MEAs. It is generally agreed that the greatest burden for the broad implementation of PB MEAs lies in the mere existence of reliable variables within administrative or other data and their acquisition for the purposes of PB MEA. Data obtained within routine healthcare may be insufficiently structured, and their availability may be subject to significant technical and legal limitations [12]. 

Further, coping with price flexibility in response to outcomes, treatment discontinuation or even product delisting in suboptimal or absent performance, and recouping payments already made to manufacturers pose a major challenge to both payers and manufacturers [5,7,13,14]. Gerkens et al. [13] highlight that the weaknesses and challenges of PB MEAs include manufacturer concerns about data sharing, regulatory and administrative costs, transferability of results and potential disincentives for providing data. Similar concerns were reported previously in our study in Slovakia [12]. Rotar [15] adds that mature HTA systems and robust IT infrastructure may be needed for some PB MEAs.

While existing studies have explored the potential of MEAs in enhancing access to medications and managing budget impact, there is limited empirical evidence on how these theoretical benefits translate into real-world settings, particularly in the context of a healthcare system like that of the Czech Republic. The nuances of legislative frameworks, stakeholder perceptions, and the actual operationalization of MEAs in such contexts are underexplored areas in current literature.

The main objective of this study is to document, analyze and clarify the expectations and concerns regarding the implementation of OB and PB MEAs within the drug reimbursement landscape in the Czech Republic. This is achieved by examining the attitudes expressed by various stakeholders and assessing the extent to which these expectations are met or unmet. Additionally, a significant part of the study involves analyzing the legislative environment, particularly the amendments of Act No. 48/1997, and how it facilitates or hinders the implementation of OB and PB MEAs. 

## 2. Materials and Methods

The literature reveals a diversity of methodologies employed in stakeholder analysis. Predominantly, stakeholder analyses incorporate both normative and instrumental approaches across various disciplines and contexts, utilizing a comprehensive array of methods. The procedural framework for stakeholder analysis typically unfolds in the following stages: 1. identification of stakeholders; 2. differentiation and categorization of stakeholders; 3. examination of the interrelationships among stakeholders and exploration of attitudes and opinions on the topic [16].

### 2.1. The Timing and Scope

This was qualitative research conducted in the period from 1 January to 10 February 2023. The semi-structured qualitative interviews were conducted with open questions (see Appendix A) among stakeholders with a professional or otherwise relevant relationship to the issue of availability, reimbursement and prices of medicines in the Czech Republic. 

It was ensured that all interviewees were familiar with the amendment of the wording of Act No. 48/1997. However, the questioning did not focus in detail on the wording of the individual provisions of this law but rather on the concept of the PB MEA and its feasibility, risks and potential benefits for the Czech Republic.

### 2.2. Identification of Stakeholders

We employed targeted snowball sampling [17] to identify stakeholders for participation in our research. All potential participants received a uniform email invitation for engagement. Selection criteria for stakeholders were predicated upon their rank and role, giving priority to senior representatives who had had a direct historical engagement in the reimbursement of drugs and medicinal products.

### 2.3. Differentiation and Categorization of Stakeholders

For our research, we decided on a stakeholder selection approach inspired by Lübbeke [18], which identifies the main stakeholders in healthcare as patients, providers (professionals and institutions), payers and policymakers (often referred to as “the four Ps” in healthcare), as well as industry, regulators, research community and media. In the first phase, a map of potential stakeholders was generated and subsequently divided into four occupational groups as a slight modification of the aforementioned framework, reflecting the specifics of the PB MEA issue. This selection was based on recognizing that each group holds unique perspectives and interests that are crucial for understanding and evaluating the possibilities and challenges associated with performance-based market entry agreements in the Czech Republic. Therefore, our methodology was carefully designed to ensure a comprehensive and balanced view of this issue. 

These groups are:Representatives of the MoH (denoted as MoH cohort),Representatives of pharma and their two local associations (Ph),Insurance companies (IC),Patients and patient organizations (PG).

Within each set, five relevant experts were selected to conduct interviews. Informed consent forms were signed by all participants.

### 2.4. Exploration of Attitudes and Opinions on the Topic

A semi-structured interview based on a predefined scenario (see Appendix A) covering key areas of the subject discussed was applied. The key areas were defined based on discussed topics on three conventions organized by the author team in the years 2021 to 2022 in the Czech Republic and Slovakia, both regarding MEA and PB/OB MEA. The key areas were:Perceived legislation about procedural, methodological and decision-making aspects of the amended Act No. 48/1997, particularly with regard to MEA;Expectations and promises related to the PB/OB MEA setting;Barriers and challenges related to the implementation of the PB/OB MEA.

The interviews were conducted and recorded online in the Microsoft Teams platform and then transcribed, ensuring even colloquialisms, jargon and vernacular expressions were captured accurately. All interviews were anonymized. MAXQDA (version 22.2.1.) was used for further processing of the transcripts. 

### 2.5. Content Analysis

Two researchers independently coded all the data. The content analysis principles were applied [19,20].

The units of analysis were defined at the level of phrases or sentences that conveyed a complete idea or concept. Each unit was categorized based on aspects central to PB MEAs, such as legal implications, stakeholder perceptions, risks, benefits and operational challenges. The researchers systematically reviewed the textual data. Each relevant segment of text was assigned a code corresponding to the predefined categories. This process was iterative, with the research team regularly discussing and resolving any discrepancies in coding to ensure reliability and consensus. 

Once coding was completed, the coded data were analyzed to identify common patterns, themes or emerging trends related to PB MEAs. This involved examining the frequency and co-occurrence of codes, understanding the context around them and interpreting the significance of these patterns in light of the research questions. The final step involved interpreting the findings from the coded data. Insights regarding the operationalization, challenges and stakeholder attitudes towards PB MEAs were integrated into the broader narrative of the study. Conclusions were drawn about the effectiveness of the current legal framework, the alignment of stakeholder interests and the potential paths for optimizing the implementation of PB MEAs in the Czech Republic.

The AI tool (ChatGPT, Version 4.0) was used in line with MDPI’s Guidelines on Artificial Intelligence and Authorship—it was used to correctly formulate ideas and conclusions without affecting the substance of the message or the content of the data.

## 3. Results

A total of 25 prominent stakeholders were approached for participation, of whom 20 agreed to be interviewed, all with demonstrable expertise in the subject matter. They were segmented into four distinct cohorts based on their affinity with the subject matter, each cohort encompassing five respondents.

Upon review by three independent researchers, a coding paradigm was obtained comprising three top clusters termed Legislation, Expectations and Promises, and Barriers and Challenges, each comprising a variable number of codes obtained from the transcript analysis. Hence a two-levels-only clustering was used.

### 3.1. Legislation

This cluster was formed by gathering abundant narratives addressing the amendment of Act No. 48/1997. A total of four codes emerged within this cluster (see Figure 1).

The code “Incompleteness” of the amendment was frequently mentioned, particularly regarding digitalization and the area of contractual policies of health insurance companies, both relevant areas linked to running and operationalizing PB/OB MEAs. The code “Insufficient specification” is perhaps not particularly surprising when it comes to legal language. It was mentioned by some respondents in a negative sense, indicating a lack of detailed specification regarding the specification of MEAs in Orphan drug regulation. On the other hand, MoH and the IC consistently expressed the view that a certain level of openness and generality in the law is appropriate, as it allows for the obligations of individual parties to be defined within the contractual relationship. This concept was even more precisely captured in the cluster “Advantages of open and general framework”. There was a broad consensus among the respondents that the legislative framework does not present any obstacles to the implementation of the PB MEA; this is reflected in the code termed “PB MEA facilitated by legislation”.

In general, MoH and ICs exhibited greater awareness and interest in the entire cluster compared to PGs. The code “PB MEA facilitated by legislation” seemed to strongly resonate with pharma stakeholders as it, obviously, could affects their business perspectives.

### 3.2. Expectations and Promises

This cluster (Figure 2) had eight codes. There seemed to be a convergent opinion on at least three attributes of PB MEA among IC, MoH and Ph. Overall, pharma articulated their perceived pros on the PB MEA more frequently than any other stakeholder group (Figure 3), well demonstrated by the code “Better and faster access to new medication”. This code was also the most frequent among other codes mentioned by representatives of the MoH and ICs. Similarly, “Fair distribution of costs and resources” was the second most abundant among those three. A similar conclusion can be retrieved from the code “Versatile benefit for all parties”. The “Greater involvement of patients” was the most frequently mentioned code by PGs.

PGs’ narratives did not overly resonate across this cluster, indicating their limited understanding of the basic principles.

Interestingly, codes “Risk sharing” and “Reduced effectiveness and budget impact uncertainty” did not resonate very frequently and certainly not among those from ICs, showing that the concept of risk and uncertainty is not seen to be attributable to the PB MEA concept. Instead, respondents frequently resorted to softer narratives more in line with the code “Versatile benefit for all parties” such as “We all have an interest in ensuring that the right patients receive appropriate treatment. Therefore, when there is a compromise that satisfies all three groups, I believe that everyone benefits from it”.

### 3.3. Barriers and Challenges

In this cluster (Figure 3), we identified five distinct codes from the recorded narratives. The code “Increased uncertainty and risk” emerged as the most frequently mentioned signal, which aligns with the overarching concept of the PB MEA. It is noteworthy, however, that stakeholders in the pharmaceutical industry emphasized uncertainty and present risk predominantly in a negative context.

The code “Lack of quality data and evidence” was strongly acknowledged by both insurance companies and pharmaceutical entities. High-quality data and evidence play a pivotal role in the successful implementation of PB MEA programs, significantly influencing the outcomes. Therefore, this code is closely linked to the previously mentioned one. Surprisingly, insurance companies possess a plethora of administrative data suitable for analyzing PB MEA or easily adaptable for such purposes. Hence, the frequent communication of this concern by insurance companies warrants attention.

The third most mentioned code, “Lack of transparency and methodology”, was notably emphasized by the Ministry of Health (MoH). However, since the interviews were conducted without specific examples, it remains unclear where a particular deficit in transparency was perceived in general. The signals captured likely pertained to the methodology and its comprehension. Stakeholders within pharmaceutical companies exhibit a notably lower level of concern regarding the absence of transparency and methodology. This could be attributed to their greater familiarity with the concept, potentially stemming from their engagement with multinational organizations.

Similarly, the category of “Roles and responsibilities” seems to be a natural extension of the preceding code, assuming that clearly defined roles and responsibilities stem from a well-established methodology. These codes collectively imply an early and immature stage of Pharmacovigilance Market Evaluation (PB MEA) in the Czech Republic. It is noteworthy that pharmaceutical companies express heightened concern about the relatively infrequent issue of “Disadvantage for patients”. Surprisingly, this concern does not appear to have substantial implications for patients themselves, nor for the Ministry of Health or payers.

## 4. Discussion

The legal landscape for PB MEAs (and, hence, OB MEAs also) in Central and Eastern Europe presents a mixed impression of readiness for this innovative reimbursement method [1]. However, it is evident that the trend towards financing highly specialized medication using models different from full reimbursement is becoming clear, with more countries allowing for innovative financing models for medicines and healthcare technologies in general. This trend aligns with observations made by other authors who have highlighted an increasing global interest in alternative financing models in healthcare [14,15]. Interestingly, this observation is consistent with the findings of Ciulla et al. [4], who noted the increasing shift towards value-based healthcare models across various healthcare systems. This shift towards value-based care underscores the necessity for systems that not only accommodate but also actively promote beneficial health outcomes and cost-efficiency, making PB MEAs particularly relevant. With the amendment of Act No. 48/1997, the Czech Republic joined the group of countries explicitly addressing MEAs in their legal frameworks, encompassing both PB MEAs.

While legal systems and terminology may vary both regionally and globally, our study has shown that in the Czech Republic, the Ministry of Health (MoH) and the Insurance Commission (IC) do not identify significant obstacles within the existing legal framework for drug reimbursement. This underscores the need for increased collaboration between pharmaceutical companies and the Czech payers.

Despite the potential transformation that this amendment could bring, it is met with a mixture of anticipation and apprehension, especially by payers who still see major barriers and challenges on the way. In this regard, insurance companies followed tightly by the Ministry of Health articulate their concerns about reliable data, evidence, transparency and methodology—four assets that, more than for any other stakeholder groups involved, are within their direct possession or under their reasonable control. Both the state-controlled payers as well as the Ministry have access to patient registries, as well as administrative data used and collected by the payers and designated organizations such as the Health Insurance Bureau or the Institute of Health Information and Statistics. Additionally, the administrative data can be enriched with so-called signal codes, which are clinically relevant pieces of information about a patient’s health status. Appropriately selected signal codes could directly serve as variables for determining treatment outcomes within Pharmacovigilance Market Evaluations (PB MEAs). Hence, it could be expected that they have the best means for the analysis and evaluation of the PB MEAs. The fact that they express the above concerns, therefore, links to their poor comprehension of the PB MEA concept or, even worse, lack of interest. The lack of interest could also be strongly supported by the expressed uncertainty and risk. 

The concept of risk-sharing or reducing uncertainty, a crucial advantage of a well-implemented Pharmacovigilance Market Evaluations (PB MEAs), does not seem to be viewed as beneficial by either party. In our assessment, this presents a substantial obstacle to fully unlocking the potential of PB MEAs, as pharmaceutical companies are likely to be hesitant to enter into agreements with potentially uncertain or adverse outcomes. Both pharmaceutical affiliates in the country and payers exhibit limited readiness to integrate uncertainty into their financial planning. This is less surprising when it comes to the payers, as they act more as administrators of healthcare budgets, driven by the inertia of the system, rather than by outcomes, thus actively managing risk. The evolving healthcare landscape necessitates robust mechanisms like PB MEAs that are adaptable to these changes, ensuring that both innovative treatments and patient care standards are not compromised. We observed a similar transformative situation in Slovakia, and surprisingly, the primary concerns were voiced more prominently by pharmaceutical companies than by regulators. This outcome suggests potential challenges to be encountered in the near future. 

While the legal framework for PB MEAs has been evolving, there remain significant mental, technical and financial hurdles that must be addressed to facilitate their widespread adoption in practice. Mentally, stakeholders in healthcare systems may require a paradigm shift in their approach to reimbursement, moving from traditional fee-for-service models to outcome-based agreements. This shift can be challenging, as it demands a reevaluation of established practices and a willingness to embrace innovative financing models and associated uncertainties. 

Financially, the implementation of PB MEAs often requires robust data infrastructure, monitoring mechanisms and investment in outcome assessment. Many healthcare systems may face budget constraints and resource limitations, making it difficult to allocate the necessary funds for these initiatives.

Addressing these mental and financial hurdles will be crucial in ensuring that PB MEAs can deliver on their promise of balancing access to innovative treatments with value for money within healthcare systems.

In conclusion, the findings suggest that while there are concerns about incompleteness and insufficient specification in the legislation, there is also a recognition of the benefits of an open and flexible legal framework. These findings reflect both the unique challenges faced in the Czech Republic and the broader global discourse on PB MEAs. Careful consideration of these issues, along with international best practices, can contribute to the effective implementation of PB MEAs in the country.

Across all key domains, we could notice a low awareness and involvement of patients and their groups in the process. This indicates a low involvement of patients in the reimbursement process, although the legislative amendment takes them into account. Greater patient involvement is desirable, but it can only be achieved if patients understand the processes sufficiently and are informed about them. In the current system, negotiations about pricing agreements, however, occur under the regime of trade secrets.

The healthcare system is a complex entity encompassing intertwined economics and the flow of information. As a system, it necessitates highly reliable information processes built on trustworthy data. In the Czech Republic, as well as in Slovakia [12], and likely in other regions, significant gaps still exist in this regard. 

In addressing the limitations and future directions of our study on Performance-Based Managed Entry Agreements (PB MEAs) in the Czech Republic, it is important to note that the scope of our qualitative research, focused within a specific timeframe and regional context, may not fully encompass the diverse experiences and opinions globally. Additionally, the confidential nature of many MEAs poses challenges in accessing comprehensive data. Future research should aim to broaden the methodological approach, perhaps integrating quantitative analysis for a more robust validation, and consider comparative and longitudinal studies to capture the evolving dynamics of MEAs in various healthcare systems. Additionally, an increased focus on patient outcomes and the impact of technological and regulatory changes could provide deeper insights into the efficiency and effectiveness of PB MEAs, guiding future policy and practice.

## 5. Conclusions

We conclude that a more comprehensive transformation is essential to enable PB MEAs to either take over or complement conventional reimbursement systems on a larger scale. The successful utilization of PB MEA is far more than the adoption of new legislation. It encompasses a cultural shift towards a more inclusive and transparent healthcare system that prioritizes patient outcomes and sustainable healthcare economics. Trust is a key quality of the process while it also poses requirements on technical infrastructure, data and availability. This transformation aligns with findings from recent studies, which underscore the importance of dynamic and flexible frameworks in PB MEAs, allowing for adjustments based on real-world outcomes and evolving healthcare needs [21]. A transformation must go beyond a singular legal act or isolated provisions related to drug reimbursement. It involves a collective effort among stakeholders to redefine and realign their roles and responsibilities towards a shared vision of healthcare excellence. It necessitates a significant redefinition of healthcare outcomes, emphasizing its value rather than treating it as a generic commodity. Incorporating lessons from various European models, as highlighted in the case studies of France and Sweden, demonstrates the potential of PB MEAs when aligned with effective data management and patient-centered approaches. This adaptability is crucial in the face of emerging healthcare challenges and rapid technological advancements that continuously reshape the landscape of health care delivery and financing. Moreover, our findings suggest that integrating patient feedback mechanisms and engaging in ongoing stakeholder dialogues are essential for the iterative development and refinement of PB MEAs. It should encompass adjustments to financial frameworks, the embrace of value-based healthcare principles—especially those concerning expenditure control policies of payers—and the facilitation of high-quality health data metrics for broader societal implementation, including health indicators. By doing so, the healthcare system can become more responsive and aligned with the needs and values of its beneficiaries, ultimately leading to improved health outcomes and system sustainability. In essence, a more holistic value-based MEA approach could provide a comprehensive framework in the future [22].

This approach encourages more discussions about improving healthcare, being responsible and ensuring everyone gets fair access to quality treatment. As we move forward, it is important that the insights from this study lead to more research and practical steps, creating a situation where PB MEAs are thoughtfully created, morally sound and commonly accepted as a standard practice.

## Figures and Tables

**Figure 1 healthcare-12-00119-f001:**
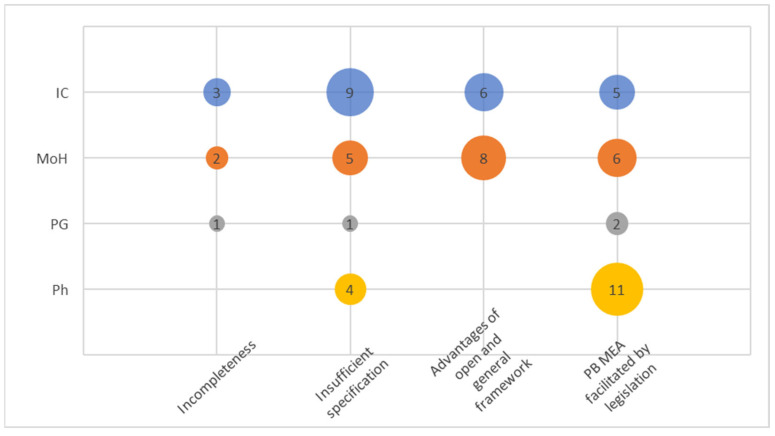
Codes within the cluster Legislation.

**Figure 2 healthcare-12-00119-f002:**
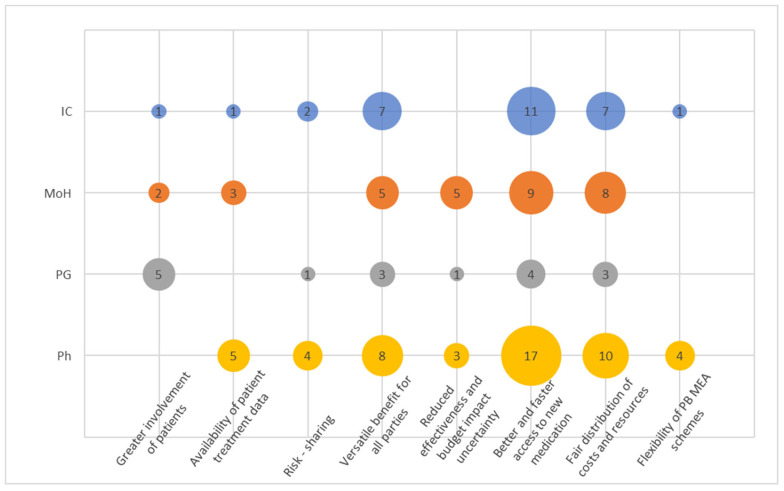
Codes within the cluster Expectations and Promises.

**Figure 3 healthcare-12-00119-f003:**
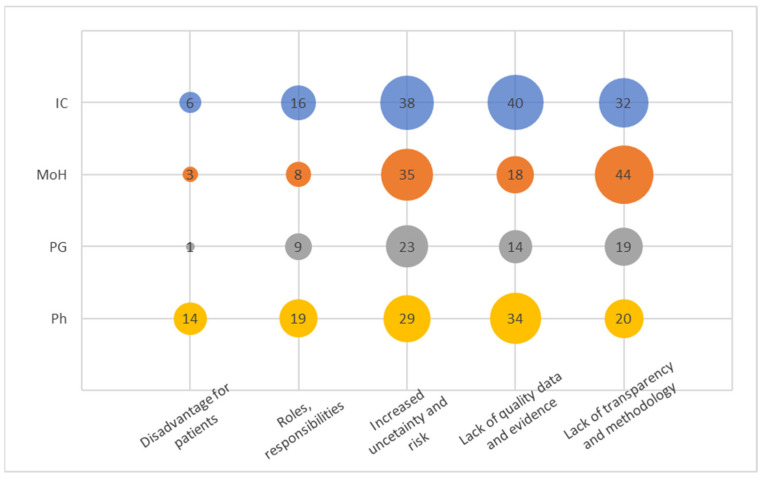
Codes within the Barriers and Challenges.

## Data Availability

Data are contained within the article or Appendix A.

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
