# Peer review of "Stakeholder Insights into Czech Performance-Based Managed Entry Agreements: Potential for Transformative Change in Pharmaceutical Access?"

_healthcare, 2024, doi:10.3390/healthcare12010119_

Round 1
Reviewer 1 Report
Comments and Suggestions for Authors
I have reviewed the paper entitled “Qualitative stakeholder research on performance-based managed entry agreements in the Czech Republic: Could a novel legal framework facilitate deep change?”. The subject of the study falls into the scope of the journal. However, there are some shortcomings in the paper. Therefore, I recommend a major revision.
1) What is the primary objective of the study? Clarify it in the introduction section.
2) Which literature gap will be filled by conducting this study? Uncover it in the introduction section.
3) Present a critical discussion of the former studies.
4) The Conclusion section is poorly written and does not provide concrete policy proposals.
5) The limitations of the study and future directions were slurred over.
6) The material and methods section was poorly written and it needs improvement in clearly presenting used methods and materials.
7) The current similarity level of the study is 16%; however, the similarity level from a single source is 5% which is above reasonable similarity levels. Besides, there is a block quotation problem in the lines between 52-72. Authors should ensure that the revised article has no more than 20% similarity in the total and 1% from a single source. Otherwise, it will require a rejection.
Comments on the Quality of English LanguageMinor revision required.
Author Response
Dear Reviewer,
Thank you for your valuable feedback and insightful suggestions on our manuscript. We have made every effort to address each of the points you raised and have incorporated changes throughout the document accordingly. We believe these modifications have significantly strengthened the paper.
| Description | Comment | Ref - corrected version |
| 1) What is the primary objective of the study? Clarify it in the introduction section. | The objectives was described in section Methods - line 87 - 92. The text was replaced and extent. | line 107 - 113 |
| 2) Which literature gap will be filled by conducting this study? Uncover it in the introduction section. | This info has been added | line 101-106 |
| 3) Present a critical discussion of the former studies. | This info has been added | line 296 -302 |
| 4) The Conclusion section is poorly written and does not provide concrete policy proposals. | It has been extent | line 38-341 and 344-346 |
| 5) The limitations of the study and future directions were slurred over. | done | line 372 to 382 |
| 6) The material and methods section was poorly written and it needs improvement in clearly presenting used methods and materials. | It has been added | line 147 - 160 and 188 - 206 |
| 7) The current similarity level of the study is 16%; however, the similarity level from a single source is 5% which is above reasonable similarity levels. Besides, there is a block quotation problem in the lines between 52-72. Authors should ensure that the revised article has no more than 20% similarity in the total and 1% from a single source. Otherwise, it will require a rejection. | corrected | link 51 -67 |
Reviewer 2 Report
Comments and Suggestions for Authors
Review report for healthcare-2766008 Qualitative stakeholder research on performance-based managed entry agreements in the Czech Republic: Could a novel legal framework facilitate deep change?
Topic
The topic should reflect the field is related to the medical field. Deep change in terms of?
Please provide relevant theories and literature review that develop this study.
Please explain why there is a need to interview these group of people.
• Representatives of the MoH (denoted as MoH cohort)
• Representatives of pharma and their two local associations (Ph),
• Insurance companies (IC),
• Patients and patient organizations (PG)
For the conclusion, please summarizing your thoughts and conveying the larger implications of your study and introducing possible new or expanded ways of thinking about the research problem. Furthermore, please change your topic to reflect the content of the study. Then, please summarise accordingly.
Comments on the Quality of English LanguagePlease avoid using too many abbreviations. It will make it difficult for the readers to read the contents smoothly.
Author Response
Dear Reviewer,
Thank you for your valuable feedback and insightful suggestions on our manuscript. We have made every effort to address each of the points you raised and have incorporated changes throughout the document accordingly. We believe these modifications have significantly strengthened the paper.
|
Comment |
Ref | |
| The topic should reflect the field is related to the medical field. Deep change in terms of? | line 2-3 | |
| Please provide relevant theories and literature review that develop this study. | ||
| Please explain why there is a need to interview this group of people. | ||
| • Representatives of the MoH (denoted as MoH cohort) | ||
| • Representatives of pharma and their two local associations (Ph), | ||
| • Insurance companies (IC), | ||
| • Patients and patient organisations (PG) | ||
| For the conclusion, please summarise your thoughts and convey the larger implications of your study and introduce possible new or expanded ways of thinking about the research problem. Furthermore, please change your topic to reflect the content of the study. Then, please summarise accordingly. | done | |
| Please avoid using too many abbreviations. It will make it difficult for the readers to read the contents smoothly. | In response to the comment regarding using abbreviations, we appreciate the reviewer's feedback. However, specific instances where the abbreviations were found to be excessive were not indicated. We believe that the information provided in parentheses serves to further specify and clarify the preceding information, thereby enhancing the reader's understanding of the subject matter. We have strived to balance the use of abbreviations with the need for clarity. We will review the text again to ensure that abbreviations are used appropriately and do not hinder the readability of the content. |
|
Reviewer 3 Report
Comments and Suggestions for Authors
The paper uses semi-structured interviews to assess stakeholder views on potential changes to a framework for managed entry agreements in the Czech Republic. The paper is clearly presented, and the methods are appropriate for this type of analysis (although of course interviews can never be considered to be at the top of the hierarchy of evidence, and the risk of bias is inevitably high).
It would have bene helpful to see a bit more discussion on the quantitative methods that can be used to inform managed entry agreements (for example, expected value of sample information).
I do not think that the paper's conclusions ("...a more comprehensive transformation is essential to enable PB MEAs to...") truly reflect the evidence (for example, line 276 reads: "...the findings suggest that while there are concerns about incompleteness and insufficient specification in the legislation, there is also a recognition of the benefits of an open and flexible legal framework.").
Finally, a minor point is that it would be helpful to include a guide to acronyms used in each of the figures. It was difficult to have to keep looking back in the text to identify what each acronym stood for.
Author Response
Dear Reviewer,
Thank you for your valuable feedback and insightful suggestions on our manuscript. We have made every effort to address each of the points you raised and have incorporated changes throughout the document accordingly. We believe these modifications have significantly strengthened the paper
| The paper uses semi-structured interviews to assess stakeholder views on potential changes to a framework for managed entry agreements in the Czech Republic. The paper is clearly presented, and the methods are appropriate for this type of analysis (although of course interviews can never be considered to be at the top of the hierarchy of evidence, and the risk of bias is inevitably high). | ||
| It would have bene helpful to see a bit more discussion on the quantitative methods that can be used to inform managed entry agreements (for example, expected value of sample information). | the chapter methods has been extended | see line 185 -204 |
| I do not think that the paper's conclusions ("...a more comprehensive transformation is essential to enable PB MEAs to...") truly reflect the evidence (for example, line 276 reads: "...the findings suggest that while there are concerns about incompleteness and insufficient specification in the legislation, there is also a recognition of the benefits of an open and flexible legal framework."). | the chapter conclusion has been corrected | see line 395 -402 and 402-414 |
| Finally, a minor point is that it would be helpful to include a guide to acronyms used in each of the figures. It was difficult to have to keep looking back in the text to identify what each acronym stood for. |
Reviewer 4 Report
Comments and Suggestions for Authors
The paper studies the qualitative stakeholders of performance-based management employment agreement in Czech Republic. It is believed that it contributes to the global discussion of PB MEAs. A more comprehensive and value-based MEA method can reshape future drug reimbursement. However, there are the following problems:
1. The method part is brief, without elaborating the specific process of the interview and the details of the detailed content analysis;
2. The number of charts is insufficient, and the information expression is not enough;
3. The references are relatively old, and the recent related research is insufficient;
4. The discussion part is a little too general, and there is no in-depth analysis of the significance of revealing the truth behind the results;
5. Some arguments in the discussion part also need to be supported by data or literature, not just the author's judgment;
6. The global discourse that contributes to PB MEAs needs to be strengthened.
Author Response
Dear Reviewer,
Thank you for your valuable feedback and insightful suggestions on our manuscript. We have made every effort to address each of the points you raised and have incorporated changes throughout the document accordingly. We believe these modifications have significantly strengthened the paper
| 1. The method part is brief, without elaborating the specific process of the interview and the details of the detailed content analysis; | done | line 145 - 160 and 185 - 204 |
| 2. The number of charts is insufficient, and the information expression is not enough; | the number of charts is corresponding towards research questions | |
| 3. The references are relatively old, and the recent related research is insufficient; | the refferences has been added | (4,11.12,20 ) from 2023 |
| 4. The discussion part is a little too general, and there is no in-depth analysis of the significance of revealing the truth behind the results; | it has been extended | line 388 - 390 and 395-402 |
| 5. Some arguments in the discussion part also need to be supported by data or literature, not just the author's judgment; | it has been extended | |
| 6. The global discourse that contributes to PB MEAs needs to be strengthened. | The discussion section has been significantly strengthened, and the conclusion part has also been notably enhanced. |
|
Round 2
Reviewer 1 Report
Comments and Suggestions for Authors
The authors have adressed most of former queries. However, similarity software reports that 7% of the study might be written by using AI tools (the lines between 359-380. Therefore, if authors have benefited AI tools, they must transparently explain details by following MDPI guideline, if not, there is a revision need in regarding section to solve AI-powered similarity.
Author Response
Dear reviewer,
thank you for the valuable feedback, I tried to incorporate everything. Information about the use of AI is in the methods chapter as well as in the acknowledgment section.
Reviewer 2 Report
Comments and Suggestions for Authors
The authors have made the necessary revisions.
Comments on the Quality of English LanguagePlease use plain and formal English so that all readers can understand the manuscript easier.
Author Response
Dear reviewer,
Thank you for the valuable feedback, I tried to incorporate everything. Information about the use of AI is in the methods chapter as well as in the acknowledgment section.